# Selectively Enhanced 3D Printing Process and Performance Analysis of Continuous Carbon Fiber Composite Material

**DOI:** 10.3390/ma12213529

**Published:** 2019-10-28

**Authors:** Huiyan Luo, Yuegang Tan, Fan Zhang, Jun Zhang, Yiwen Tu, Kunteng Cui

**Affiliations:** Institute of Advanced Materials and Manufacturing Technology, Wuhan University of Technology, Wuhan 430070, China; lhy_lw430@whut.edu.cn (H.L.); ygtan@whut.edu.cn (Y.T.); 13657201521@163.com (J.Z.); ywtu_whut@163.com (Y.T.); ckt@whut.edu.cn (K.C.)

**Keywords:** continuous carbon fiber composite, 3D printing, volume fraction, selective enhancement

## Abstract

Aiming at the limited mechanical properties of general thermoplastic 3D printed models, a 3D printing process method for selective enhancement of continuous carbon fiber composite material is proposed. Firstly, the selective enhanced double nozzle working mechanism and crafts planning process are put forward. Then, based on the double nozzle carbon fiber 3D printing device, test samples are printed by polylactic acid (PLA) and carbon fiber material, and the test samples are enhanced by inserting layers of continuous carbon fiber material. The performance test of the samples is carried out. Experiment results show that when the volume fraction of continuous carbon fiber material increases gradually from 5% to 40%, the tensile strength increases from 51.22 MPa to 143.11 MPa. The performance improvement curve is fitted through experimental data. Finally, field scanning electron microscopy is used to observe the microscopic distribution of continuous fibers in the samples. The results of the research lay the foundation for the performance planning of 3D printed models.

## 1. Introduction

In the 30 years of the development of additive manufacturing technology, 3D printing has gradually broken through the product design concepts for manufacturing process and achieved the product design concepts for the performance of products [1]. This technology is widely used in automobile manufacturing, aerospace, military, medical, construction, education and scientific research [2,3,4]. Carbon fiber (CF) is a new type of 3D printing material. The CF 3D printing products can have excellent performance and light weight characteristics compared to metal materials [5,6]. The traditional manufacture of carbon fiber models is mostly formed by different molds under a certain temperature and pressure. In this way, the manufacturing cost is high and the time is long [7,8]. However, carbon fiber materials combined with 3D printing technology can form complex models at a low cost and high efficiency. According to traditional thermoplastic material 3D printing, it has the characteristics of low cost and easily used equipment which has been widely used in modeling design and verification [9,10]. However, it is difficult to satisfy the performance requirements by the traditional thermoplastic materials 3D printing method [11]. Although the pure continuous carbon fiber 3D printing models have better performance, this method has weaknesses, such as higher cost, longer molding time, and a number of key technologies are still being further studied. Therefore, the 3D printing process reported here, which uses continuous carbon fiber composite material to selectively enhance the thermoplastic model, can combine the advantages of both the above and further expand the application range of 3D printing.

At present, the research on 3D printing of carbon fiber reinforced composite materials in the world includes two aspects: the short fiber reinforced and the long fiber reinforced 3D printing [12]. For short fiber reinforcement, short-cut carbon fibers of different lengths and contents are mixed with other printing materials. Then, they are put into the extruder to obtain a 3D printing filament. This filament finally is used to print products and experimental samples [13,14]. Karsli et al. prepared carbon fiber reinforced polyamide 6 (PA6) composites by using melt mixing method. Mechanical test results showed that increasing CF content increased the tensile strength, modulus and hardness values [15]. 

However, the reinforced effect of the short fiber is far less than the long fiber. Bade et al. prepared a pure polylactic acid (PLA) tensile test model and a PLA/CF tensile test model reinforced by carbon fiber bundle through melt molding. Comparing the tensile performance test of the above two models, the tensile strength of the carbon fiber reinforced experimental sample increased by 73% compared to the pure PLA experimental sample [16]. Xia created an additive manufacturing device for continuous fiber reinforced composite material by fusing resin and fiber materials together and extruding them from the nozzle. He explored the effects of printing speed, temperature, fiber pre-tightening force on the shape quality of the formed sample; fiber neutrality, tensile strength and bending strength of the formed sample [17]. Matsuzaki et al. introduced a 3D printing method for continuous fiber reinforced thermoplastics based on fused deposition modeling. The carbon fiber material and PLA material were separately fed. Then, they were heated and mixed in the nozzle. Compared to the PLA 3D printing sample without fiber material, the properties of the models which contained these two materials considerably improved [18]. Mori et al. studied this method and first manufactured a lower plate and upper plate by fused deposition modeling (FDM), thereafter, CCF was sandwiched between both plates; and finally, the three parts were bonded by thermal treatment. The results showed that the strength increased to almost double of the previous values by using this method [19]. Hu et al. proposed a printing method for continuous carbon fiber composite models by modifying the 3D printer extrusion head to achieve long fiber bundle printing. After testing, they discovered that the flexural strength and flexural modulus of printed composites significantly improved with the proposed method with specified printing parameters, and the layer thickness had the greatest contribution towards the final flexural strength [20]. Tian et al. proposed a method for preparing continuous fiber reinforced thermoplastic composite (CFRTPC) material based on 3D printing technology. Continuous carbon fiber and PLA filaments were used as the reinforced phase and matrix respectively, and they were simultaneously extruded by FDM 3D printing process to achieve integrated preparation and the forming of composite materials [21]. Pruβ et al. designed a new type of 3D printer head able to transport fiber bundle on both sides of the original machine head to achieve continuous fiber reinforced printing [22]. Brooks and Molony adopted another methodology which was to design channels within the parts that may be filled with continuous reinforcement [23]. Yang et al. designed a continuous fiber reinforced thermoplastic composite extrusion head. Then, they made composite material samples and performed multiple mechanical experiments. The results showed that continuous fibers had a considerable degree of enhancement on the flexural strength and tensile strength of thermoplastic materials [24]. All the above 3D printing research for long fiber materials made structural modifications or designs for a single nozzle. The carbon fiber composite material could not be selectively filled in the model at the time of printing. In this way, the cost of forming a model is high. Markforged released the desktop carbon fiber 3D printer Mark One. One of the nozzles extruded the nylon material and the other nozzle extruded the special fiber reinforced material [25]. However, the method of reinforced material filling was a single contour filling pattern. In summary, the research for long-fiber 3D printing in the world is mainly based on the printing of single-filament bundle carbon fiber. However, in this way, the forming time is long, the printing cost is high, and the model cannot be selectively enhanced. Therefore, the dual-nozzle FDM 3D printing technology for thermoplastic material and continuous carbon fiber composite material is studied here. A new, selectively enhanced method is proposed, which can be applied to different reinforced materials. Different layers can be flexibly selected to print the reinforced material, and the appropriate filling method can be selected according to the material properties. Through this selective insertion of continuous carbon fiber composite material into thermoplastic material, the specific enhancement method is proposed and the performance test is designed. 

## 2. Continuous Carbon Fiber Reinforced 3D Printing Process

### 2.1. Materials

PLA has been widely used in many thermoplastic 3D printing applications. However, the mechanical properties of models printed by PLA are limited. Due to this reason, PLA is chosen in this study as the matrix material to be reinforced by continuous carbon fibers. The PLA 4032D particles material comes from American Nature Works and the carbon fiber material (T700) comes from Toray Japan. The data of two materials are shown in Table 1. The dry carbon fiber tow is firstly let to pass through the entire preparation device. Then, the PLA particles are placed in a closed container and heated to a molten state at 210 °C. The two materials are mixed at the same temperature. They will then pass through a nozzle of 0.6 mm at a speed of 10 mm/s. They are finally cooled in normal room (25 °C) temperature water to form continuous carbon fiber PLA composite material (black filament in Figure 1). The whole material preparation process is shown in Figure 2. 

### 2.2. Methods

Through the proposed method of printing thermoplastic material and carbon fiber composite material by double nozzle, the enhanced properties of carbon fiber for thermoplastic materials has been achieved. As shown in Figure 3a, continuous carbon fiber composite materials (black filament) and thermoplastic materials (white filament) are extruded into the respective nozzles by extrusion device. The surface resin of the continuous carbon fiber composite is melted by heating with two heating rods at the nozzle. This allows the layer of continuous carbon fiber material to be bonded to the layer of thermoplastic material. During the printing process, according to the planning of different types of material for different layers, the corresponding nozzles are selected to print. By inserting different layers of continuous carbon fiber composite material into the thermoplastic material model, the thermoplastic material model can be selectively enhanced. Therefore, in order to complete the above selectively enhanced process, it is necessary to achieve the printing method of carbon fiber reinforced material for thermoplastics and the specific planning method. 

The two nozzles are rigidly connected as Figure 3b,c shows. These are the nozzle 1 extrude continuous carbon fiber composite material and the nozzle 1 extrude thermoplastic material. When one nozzle extrudes, the other nozzle stops extruding. As the carbon fiber reinforced material nozzle is required to print the reinforced material on the top of the corresponding thermoplastic nozzle printed layer, it is necessary to set the positional relationship of the two nozzles. Thereby, the carbon fiber reinforced nozzle can print the carbon fiber reinforced material in the corresponding layer number and position to achieve selective enhancement in one model.

Since the nozzle moves to the print point in the G code file, the nozzle 1 is the original position of the coordinate system, and the distance between the two nozzles needs to be set when the nozzle is switched. The whole nozzle position conversion is shown in Figure 4. Assuming that the origin of the printing platform is (0, 0, 0), the position of the nozzle 1 is ($x_{1}$, $y_{1}$, $z_{1}$), and assuming that the distance of the nozzle 2 relative to the nozzle 1 is (Δx, Δy, Δz), then the position of the nozzle 2 in the printing platform is ($x_{1}+\Delta x$,$y_{1}+\Delta y$,$z_{1}+\Delta z$). When the nozzle 1 moves to point A (a, b, c) in the coordinate system with the nozzle 1 as the origin, the position of the nozzle 1 in the printing platform becomes ($x_{1}+a$, $y_{1}+b$, $z_{1}+c$). When the nozzle 2 is also moved to point A, nozzle 2 in the printing platform also becomes ($x_{1}+a$, $y_{1}+b$, $z_{1}+c$). The fixed offset distance Δx, Δy, Δz between the two nozzles is decided by the printer head design, and Δz is 0.

The 3D printing planning process is shown in the Figure 5. First, the models that need to be printed require three-dimensional modeling. After the modeling is completed, the STL file needs to be generated. Then the STL file needs to be processed by the slicing software. The positional distance between the nozzles need to be set first. The number and the position of the layers which need to be printed with the carbon fiber composite material need to be set after that. The filling pattern that suits for the carbon fiber composite material needs to be set, such that the G code file is generated layer by layer. Based on the G code, the entire model is then 3D printed, containing the corresponding carbon fiber selectively reinforced layers. Current commercially available continuous carbon 3D printing technology requires proprietary slicing software to print a model [28]. This software is “closed source” and does not allow for user adjustment of key printing parameters such as temperature, nozzle movement or extrusion speed. This limits the printing capabilities as the printing settings cannot be fully customized [28]. Moreover, for the deposition of carbon fibers, only a circumferential fill pattern is possible which fills the shape from the outside inward in a spiraling motion. This means the fiber is always orientated along the outer perimeter of the part. In contrast to this, the selectively enhanced method reported here, allows for the number and the position of layers for carbon fiber to be flexibly chosen, and for the content of carbon fiber in the reinforcing layer to be adjusted by changing the density of this layer. In here, the fiber content is determined by setting the path filling density. When the filling path density is higher, it means the distance between the two path lines is shorter, so the filling path is denser and filled fibers are more, and the content of carbon fiber is higher. Otherwise, when the filling path density is lower, the content of carbon fiber is lower. The filling pattern for carbon can also be chosen.

## 3. Performance Testing of Carbon Fiber Selective Reinforced Models 

Different layers of continuous carbon fiber composite material will be selectively inserted to the model during printing by executing the pre-generated G code command. Through the tensile test, we will test the reinforced effect of different volume fractions of continuous carbon fiber composite material for the PLA model test models. 

### 3.1. Continuous Fiber Reinforced Layer Path Selection

For this, dumbbell type test specimens are 3D printed. The filling patterns that can be selected are mesh filling, line filling and spiral offset filling as shown in Figure 6.

The red parts in the above figures are the nozzle jump parts. As the composite material is continuously fiber reinforced, the material needs to be cut. The nozzle jump points means that the printed head will stop extruding the materials and move to the next print point. Since the extruding behavior of carbon fiber composite material is different compared to thermoplastic material, it cannot be pumped back when intercepting the composite filament printing, which means it is necessary to cut the fiber in advance when the nozzle jumps. Otherwise, the fiber will print in the current layer along the nozzle path when the nozzle moves. The shape and the properties will be affected by the results caused by nozzle jump points. Comparing the three filling patterns, it can be seen that when the mesh filling pattern is chosen, there are many jump points during the model printing, which means that this filling pattern is not conducive to print continuous long carbon fiber composite material. At the same time, the filling direction of this filling pattern is not aligned with the central axis direction of the model, which combined with the anisotropy of the carbon fiber composite materials will cause reduction of the testing specimen tensile strength. When the line filling and spiral offset filling patterns are used, the filling direction of the middle stretching section is along the central axis direction, and this can effectively enhance the tensile performance of the test specimens. However, there are multiple jump points in the line filling pattern, while there is only one jump point in the spiral offset filling pattern. Due to this, the spiral offset pattern is more suitable for printing of continuous long carbon fiber composite materials. After the filling pattern has been selected, the model diagram and dimension drawing of the models with continuous carbon fiber composite material inserted are shown in Figure 7.

### 3.2. Performance Test of Carbon Fiber Reinforced Models

The model is set to 5 mm and the layer thickness is 0.25 mm, such that the total number of layers per sample is 20. The middle 0 to 8 layers of test models are selected to use continuous carbon fiber PLA composite for printing. The remaining layers are selected to use PLA material. For example, we can choose to insert 1 layer of continuous carbon fiber composite material in the 10th layer, insert 2 layers of continuous carbon fiber composite material in the 10th and 11th layers, and insert 3 layers of continuous carbon fiber composite material in the 9th, 10th and 11th layers and so on. 8 layers of continuous carbon fiber composite material are inserted into the models, the proportion of the number layers of carbon fiber material to the total 20 layers is 0%, 5%, 10%, 15%, 20%, 25%, 30%, 35%, 40%, respectively. The specific insertion method is shown in Table 2. In this way, the continuous carbon fiber composite material with different volume fractions can be inserted into the test models as shown in Figure 8 (Figure 8a shows the spiral offset filling pattern, also shown in Figure 6c), where this first layer has not been printed completely. After the outer area in Figure 8a has been printed, the fiber is cut and then the inner area in Figure 8b is printed. In this printing pattern, due to the anisotropy of carbon fiber, the carbon fiber filling direction in the middle of this model can be filled along the central axis direction of the model. Through tensile experiment for test models, the effect of the content of continuous carbon fiber composite material on the properties of models can be evaluated.

In Figure 8, the 3D printing quality may not print very well. In (a), some right-angled part is printed like a rounded shape. This is because the fibers on the front right-angled side are not completely bonded to the PLA when printing another right-angled side. In (b) and (c), there exists some featuring filament waviness. These are because the front printed fiber layer is not flat enough. So, the rapid bonding of fiber materials and PLA, and the flatness of the fiber layer will have an important influence on model printing. The above tensile test models were tested for mechanical properties on a SmartTest universal testing machine. The geometry and dimensions of the test specimens are shown in Figure 9. The tensile loading speed was 5 mm/min. The tensile strength value of the test samples was recorded. The typical test specimen and tensile test machine gripping are shown in Figure 9.

### 3.3. Analysis of Performance Results

The elastic modulus and tensile strength of the test models which are inserted in 0 to 8 layers of carbon fiber composite material, and the remaining layers of PLA are presented in Table 2.

The corresponding stress-strain curves are shown in Figure 10. It can be seen that the tensile strength of the pure PLA test model is 36.89 MPa. For the test model in which 1 layer of continuous carbon fiber composite material is inserted, the number of continuous carbon fiber composite layers accounts for about 5% of the total number of layers, and tensile strength reaches 51.22 MPa, which is 38.8% higher than that of the pure PLA test sample. For the test sample in which 8 layers of carbon fiber composite materials were inserted, the number of continuous carbon fiber composite layers accounts for about 40% of the total number of layers, and the tensile strength reaches 143.11 MPa, which is 287.9% higher than that of the pure PLA test sample. The modulus gradually increases when the number of CF layers increases, as shown in Figure 11. The strength change trend of the test model, in which the continuous carbon fiber composite material is inserted, is shown in Figure 12. The strength increase fitting curve is y = 12.02x + 37.49. The tensile strength of the pure carbon fiber composite material test samples is 291.575 MPa as shown in Figure 13. The result of the 20-layers test model calculated by the fitting curve is 277.89 MPa, which is in good agreement with the experimental value in Figure 13.

The test samples of 2 and 4 layers of CF are analyzed through field emission scanning electron microscopy with model JEM-7500F and shown in Figure 14. It can be seen that the carbon fiber material in the SEM image, in which the 4-layers carbon fiber composite material is inserted, is denser than the carbon fiber material in the SEM image in which the 2-layers carbon fiber composite material is inserted, and the degree of dispersion of the carbon fiber material is low.

## 4. Conclusions

In existing research (as [17,18,19,20,21,22,23,24] mentioned), the research for long-fiber 3D printing in the world is mainly based on the printing of single nozzle. In this study, the dual-nozzle FDM technology is developed and proposed for 3D printing of continuous carbon fiber composite and thermoplastic materials. Taking the advantage of the excellent mechanical properties of carbon fiber materials, the carbon fiber selective insertion for the model has been achieved. By inserting different layers of carbon fiber composite materials into the test samples, the model contains different volume fractions of carbon fiber composite materials. When the volume fraction accounts for 40%, the tensile strength is increased by 287.9% compared to the pure PLA model. Compared with the existing data, this promotion radio is competitive (e.g., performance improvement ratio is 185.7% [29]). The relationship between carbon fiber reinforcement ratio, enhancement materials and the tensile strength of the final 3D printed material is proposed based on fitting the experimental data. Besides, current commercially available continuous carbon 3D printing technology (Markforged) does not allow the user to change key printing parameters such as temperature, nozzle movement or extrusion speed (as [28] mentioned), but for this selectively enhanced method, the number and the position of carbon fiber layers can be user chosen, and the content of carbon fiber in this layer can be edited by changing the density of this layer. The filling pattern for carbon can also be chosen. This allows different parameter settings and path selection for different models. This 3D printing selective reinforcement method is more flexible compared to existing commercially available similar technology.

## Figures and Tables

**Figure 1 materials-12-03529-f001:**
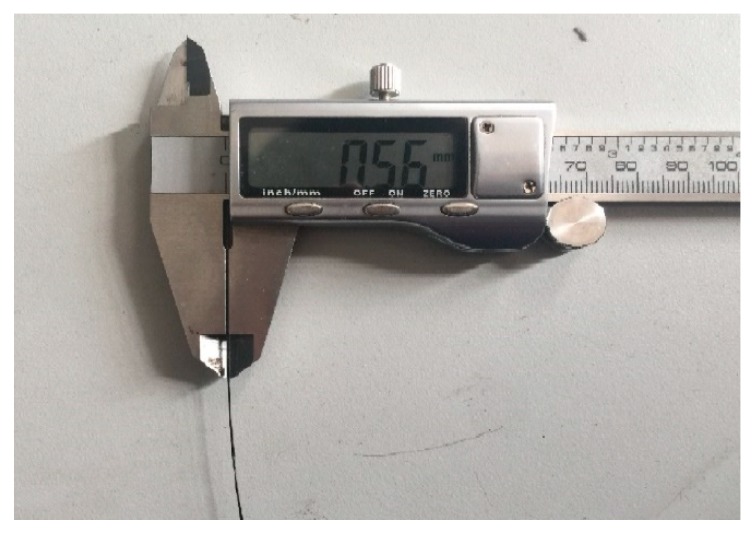
Carbon fiber material.

**Figure 2 materials-12-03529-f002:**
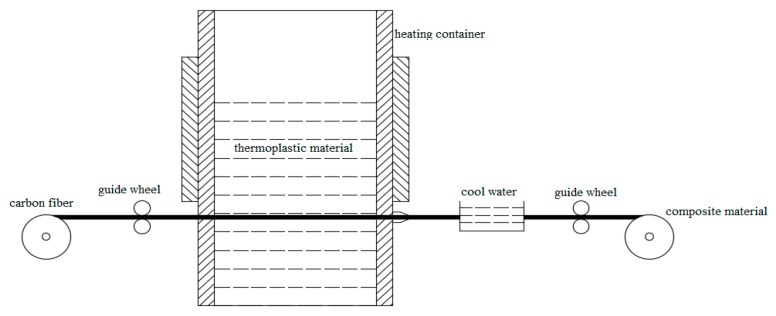
Material preparation process.

**Figure 3 materials-12-03529-f003:**
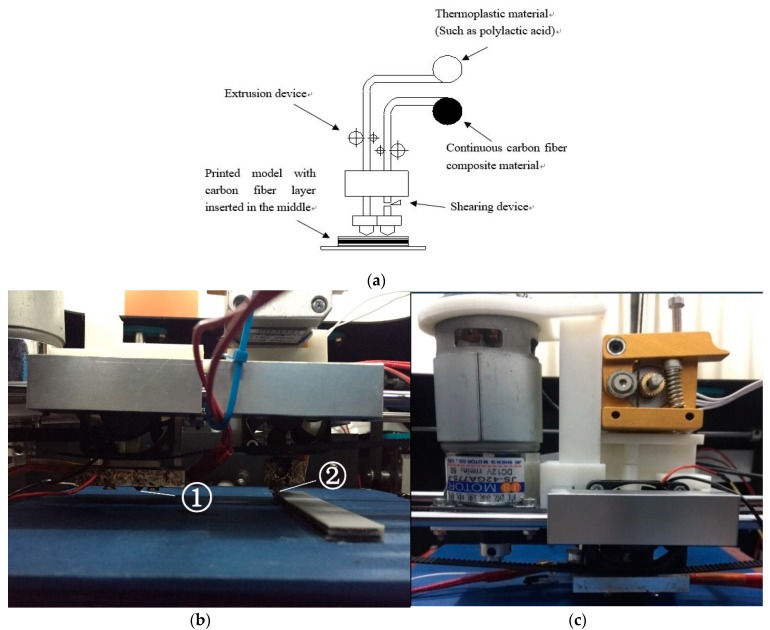
Dual nozzle 3D printing process and 3D printer head: (**a**) Schematic diagram of dual nozzle 3D printing process; (**b**) 3D printer dual head; and (**c**) the printer head for carbon fiber composite material.

**Figure 4 materials-12-03529-f004:**
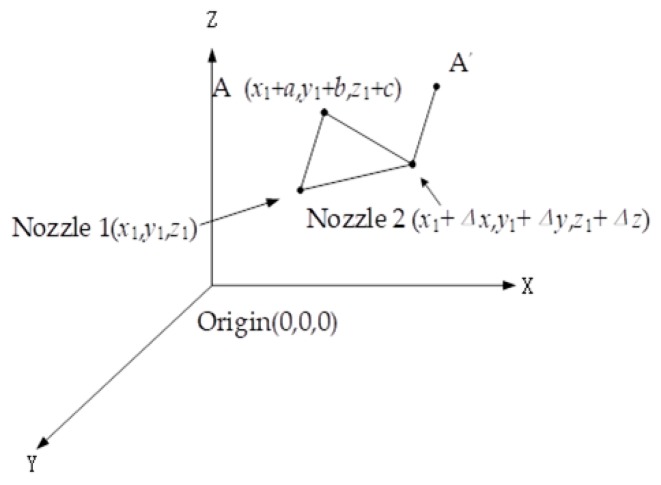
Nozzle position conversion.

**Figure 5 materials-12-03529-f005:**
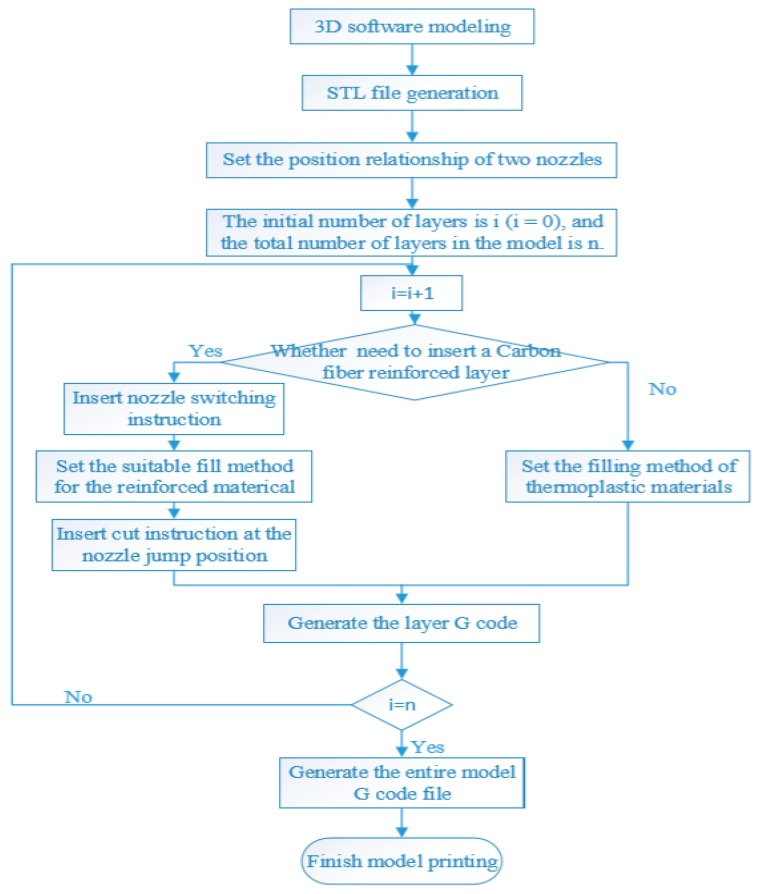
3D printing planning process.

**Figure 6 materials-12-03529-f006:**
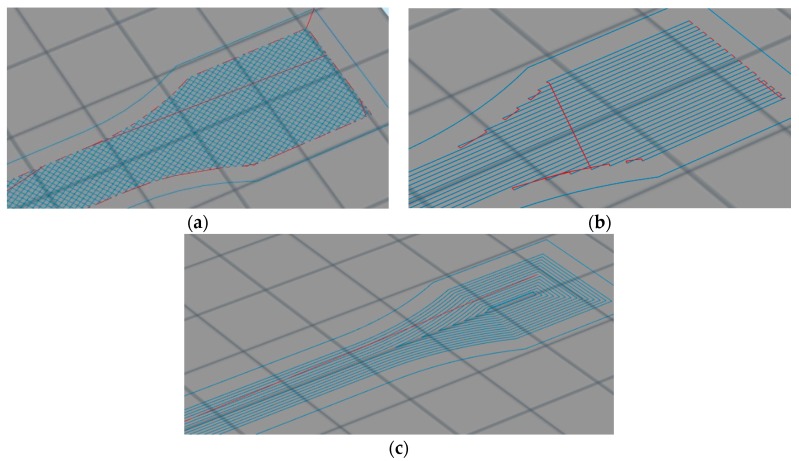
Possible filling patterns: (**a**) mesh filling; (**b**) line filling; and (**c**) spiral offset filling.

**Figure 7 materials-12-03529-f007:**
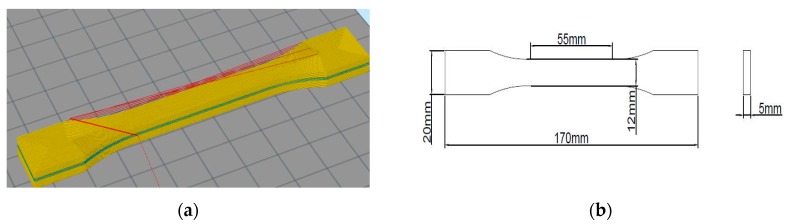
(**a**) Whole test sample model; and (**b**) model size.

**Figure 8 materials-12-03529-f008:**
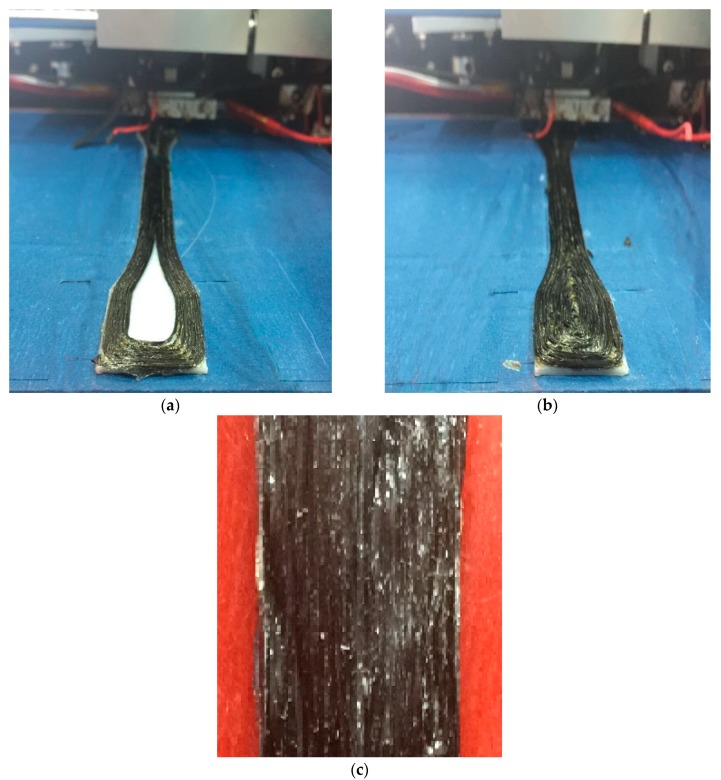
Printing process: (**a**) outer area printing process; (**b**) whole layer printing process; and (**c**) middle area of the model.

**Figure 9 materials-12-03529-f009:**
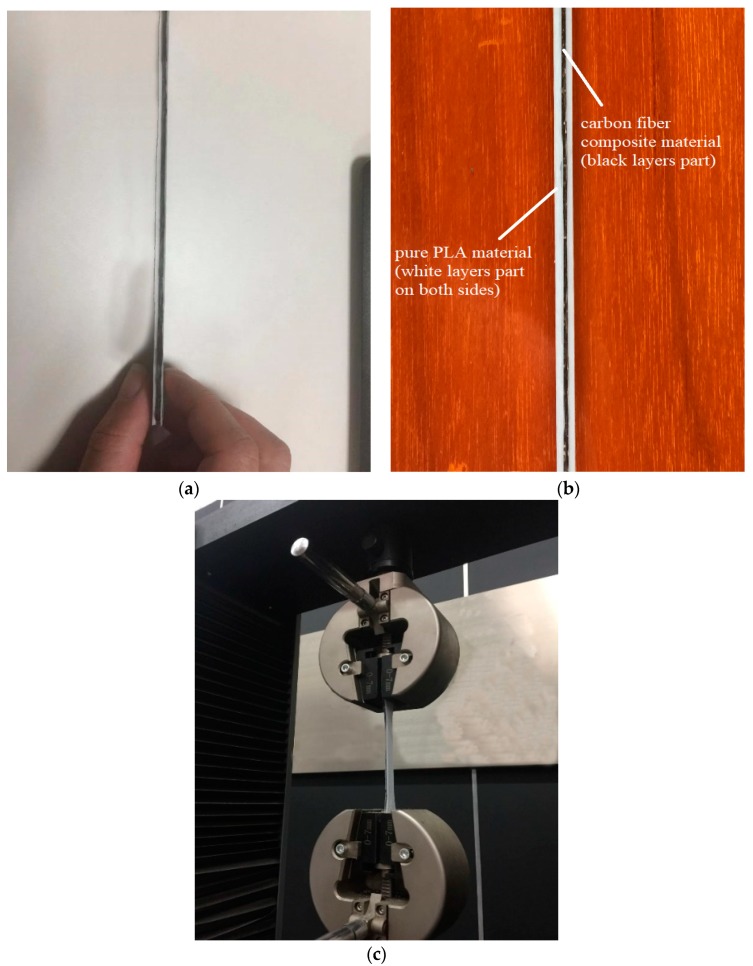
Test model: (**a**), (**b**) the typical test specimen; and (**c**) tensile test machine gripping.

**Figure 10 materials-12-03529-f010:**
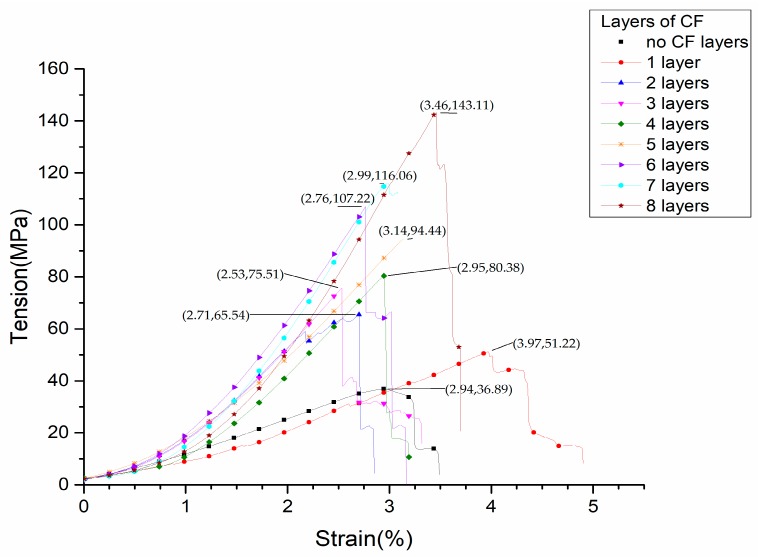
Stress-strain curves of selectively reinforced 3D printed specimens.

**Figure 11 materials-12-03529-f011:**
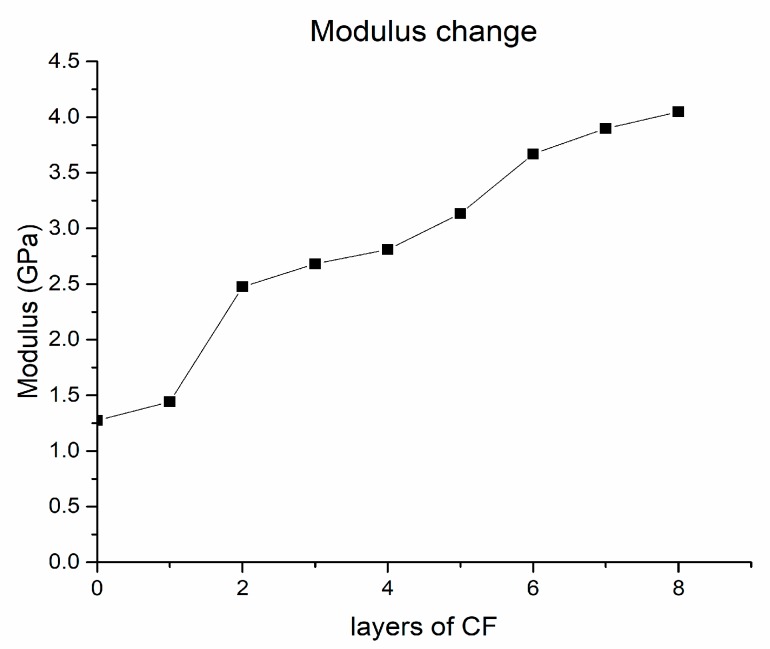
Modulus change.

**Figure 12 materials-12-03529-f012:**
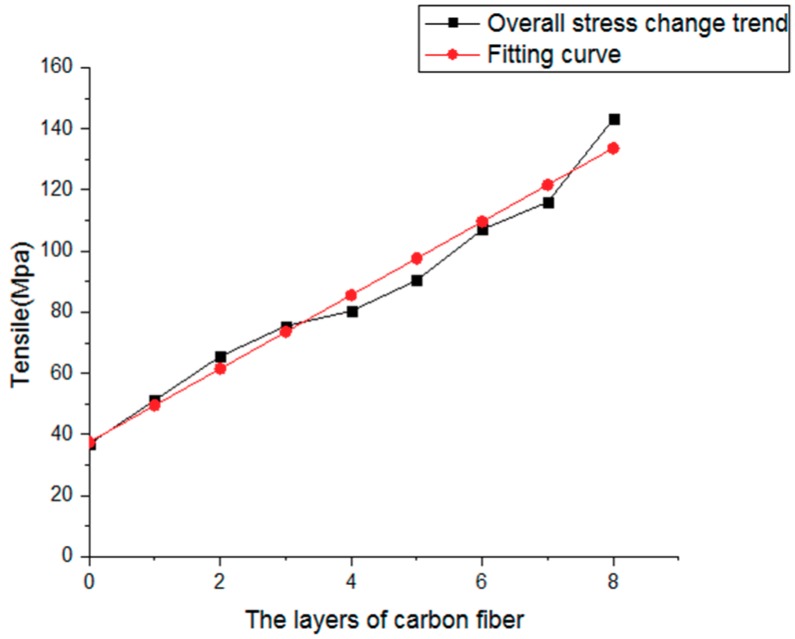
Overall stress trend change.

**Figure 13 materials-12-03529-f013:**
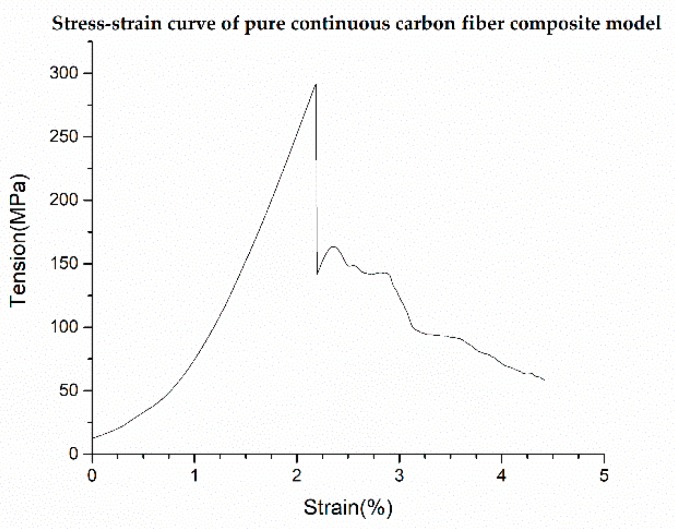
Stress-strain curve.

**Figure 14 materials-12-03529-f014:**
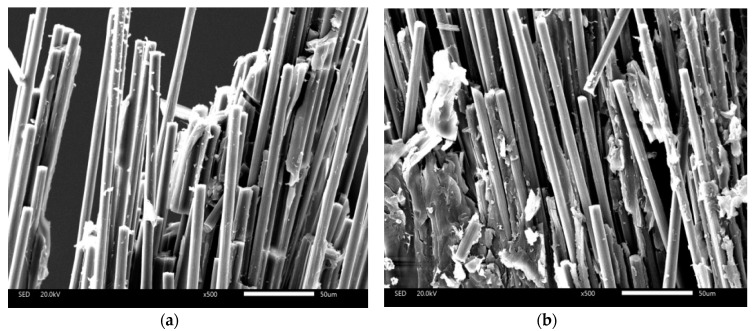
SEM image is 500 magnifications and the staff gauge is 50 μm. (**a**) SEM of 2 layers of carbon fiber; and (**b**) SEM of 4 layers of carbon fiber.

**Table 1 materials-12-03529-t001:** The data sheets of PLA (polylactic acid) particles and carbon fiber material.

Mechanical Property	PLA [26]	Carbon Fiber [27]
Density/ ($g/{\mathrm{cm}}^{3})$	1.24	1.8
Melting Point/ (°C)	155~230	
Fracture Elongation/ (%)	7.0	2.1
Tensile Strength/ (MPa)	53	4900
Tensile Modulus/ (GPa)	3.5	230
Rockwell Hardness	88	

**Table 2 materials-12-03529-t002:** Continuous carbon fiber composite material selective insertion layer.

Total Layers	Amount of Carbon Fiber Layers	Carbon Fiber Volume Fraction (%)	Position of the Inserted Layers	Tensile Strength of Models (MPa)	Modulus (GPa)
20	0	0	No inserting	36.89	1.27
20	1	5	10	51.22	1.44
20	2	10	10–11	65.54	2.48
20	3	15	9–11	75.51	2.68
20	4	20	9–12	80.38	2.81
20	5	25	8–12	94.44	3.13
20	6	30	8–13	107.02	3.67
20	7	35	7–13	116.06	3.90
20	8	40	7–14	143.11	4.05

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
