# Peer review of "Selectively Enhanced 3D Printing Process and Performance Analysis of Continuous Carbon Fiber Composite Material"

_materials, 2019, doi:10.3390/ma12213529_

Round 1

Reviewer 1 Report

3D printing of PLA composites filled with different fillers are widely reported. Therefore, the materials dealt in this manuscript isn't novel. However, designing of the 3D printed materials for UTM measurement will stimulate researches on mechanical and physical properties of the 3D printable polymers. Thus, I recommend this manuscript to be accepted in the Materials journal in the present form.

Reviewer 2 Report

As a reviewer I have to propose to reject the paper because the findings are incremental and the manuscript has insufficient contribution to existing knowledge. Especially, it is unclear what is the scientific insight emerging from this study. Also, the manuscript in general reads as a technical report rather than a scientific paper and it is difficult to follow due to the poor English.

Also, it is not clear how different layers of the continuous carbon fibre are selectively inserted to the model during printing.

Very low quality images such as Fig. 1, Fig. 3 and Fig. 13 (Fig. 2 is missing) that makes it difficult to read.

The review of the literature is insufficient and needs to be extended an updated.

This clearly indicates that the paper lacks novelty and accordingly I would suggest that the authors consider reworking the paper for a future submission with a focused problem statement and work to probe novel research findings that would help for an acceptable paper. It is unfortunate that I have to be negative but hopefully the comments will be interpreted constructively.

Reviewer 3 Report

The manuscript deals with 3D printability of carbon fibre-PLA composites and their mechanical properties. A major revision is needed based on the following comments.

1.      Why PLA? Explain the novelty factor of the work, which is missing from the introduction.

2.      What does this dual nozzle printing advantage over single nozzle printing?

3.      Why 20 layers in total and why only up to 8 layers of CF?

4.      Stress-stain curves are not quite visible. Enlarge or present it clearly.

5.      Discussing about tensile strength. What about modulus? How does it change?

6.      SEM – no scale is visible. It is of poor quality with high contrast.

7.      Way too many Figure numbers. They should be reduced my merging relevant figures under a single caption. From 4 to 12.

8.      Figure 11 is practically useless.

9.      Figure 12- stretching should be focused on. The current fig is poorly presented.

10.  The manuscript has significant language flaws (grammar, typos, punctuations, use of capital letters, and abrupt changes of tenses). Should be given more attention.

11.  There are redundant sentences like “The stress (tensile strength) results are summarized as follows:”

12.  Basic errors:

a.      Space between unit and value is not consistent throughout.

b.      Mpa should be MPa

c.       Very few references have been cited -  should be more.

Round 2

Reviewer 2 Report

The reviewer went through the revised manuscript carefully trying to detect changes and improvements, however, the reported work still lacks novelty and has insufficient contribution to existing knowledge. Looking at the drawn conclusion, for example, one can clearly see that the finding of this research work is very well known, "the strength of the 3D printed sample is highly dependent on the volume fraction of carbon fiber composites" particularly, the higher the volume fraction of carbon fiber inserted, the more strength printed parts can be achieved. The author should focus on the proposed approach to selectively inserting the carbon fiber to improve the strength performance of the printed part. However, even though, this proposed method needs improvement to be considered for publication. For example to selectively insert carbon fiber layer in complex real 3D printed parts to enhance the performance in predefined locations/areas. Accordingly, I would suggest that the authors consider reworking the paper for a future submission with a focused problem statement and work to probe novel research findings that would help for an acceptable paper. It is unfortunate that I have to be negative again but hopefully the comments will be interpreted constructively. 

Author Response

Response to Reviewer 2 Comments

Thank you for your valuable comments.

  This article has been revised and reorganized, and is more biased towards selective enhancement methods.According to MarkOne printer, to print a model, a proprietary slicing software must be used. This software is “closed source” and does not allow for user adjustment of key printing parameters such as temperature, nozzle movement or extrusion speed. This limits the printing capabilities as the printing settings cannot be fully customized. For the deposition of carbon fibers, only a circumferential fill pattern is possible which fills the shape from the outside inward in a spiraling motion. This means the fiber is always orientated along the outer perimeter of the part. But for this selectively enhanced method, the number and the position of layers for carbon fiber printing can be chosen and the content of carbon fiber in this layer can be edited by changing the density of this layer. The filling pattern for carbon can also be chosen. As for selectively insert carbon fiber layer in complex real 3D printed parts to enhance the performance in predefined locations/areas, we are now working on reinforcement fills for flanged shafts and gears. The specific enhancement effect on these components is also the direction of our future research.

Reviewer 3 Report

The revised manuscript is improved. Please follow the comments below to address some of the minor issues.

Stress-strain graphs are too clumsy. Needs clearer presentation as it is difficult to follow the colour choices. Use of capital letter and space between words are having some issues. Please carefully check the manuscript throughout. The capital letter should not come in between. Figure 5 and 6 can be under a single caption. Figure 6 – are the dimensions in ‘mm’?

Author Response

Response to Reviewer 3 Comments

First thank you for your valuable comments

Point 1: Stress-strain graphs are too clumsy. Needs clearer presentation as it is difficult to follow the colour choices.

Response 1: Added graphic markers to the figure, not just colors.

Point 2: Use of capital letter and space between words are having some issues. Please carefully check the manuscript throughout. The capital letter should not come in between.

Response 2: These mistakes have been revised.

Point 3: Figure 5 and 6 can be under a single caption. Figure 6 – are the dimensions in ‘mm’?

Response 3: They are under a single caption now, and the dimensions are in ‘mm’.